# Characterization of Carbapenemase-Producing *Klebsiella pneumoniae* Isolates from Two Romanian Hospitals Co-Presenting Resistance and Heteroresistance to Colistin

**DOI:** 10.3390/antibiotics11091171

**Published:** 2022-08-30

**Authors:** Annamária Főldes, Mihaela Oprea, Edit Székely, Codruța-Romanița Usein, Minodora Dobreanu

**Affiliations:** 1Department of Microbiology, Laboratory of Medical Analysis, “Dr. Constantin Opriș” County Emergency Hospital, 430031 Baia Mare, Romania; 2Doctoral School of Medicine and Pharmacy, “George Emil Palade” University of Medicine, Pharmacy, Science and Technology, 540142 Târgu Mureș, Romania; 3“Cantacuzino” National Military Medical Institute for Research and Development, 103 Splaiul Independenței, 050096 Bucharest, Romania; 4Department of Microbiology, Central Clinical Laboratory, County Emergency Clinical Hospital, 540136 Târgu Mureș, Romania; 5Department of Microbiology, “George Emil Palade” University of Medicine, Pharmacy, Science and Technology, 540142 Târgu Mureș, Romania; 6Department of Clinical Biochemistry, Central Clinical Laboratory, County Emergency Clinical Hospital, 540136 Târgu Mureș, Romania; 7Department of Laboratory Medicine, “George Emil Palade” University of Medicine, Pharmacy, Science and Technology, 540142 Târgu Mureș, Romania

**Keywords:** carbapenemase-producing Enterobacterales, Vitek 2 Compact, broth microdilution, whole-genome sequencing, antimicrobial resistance, colistin resistance, colistin heteroresistance

## Abstract

*Klebsiella pneumoniae* is a notorious human pathogen involved in healthcare-associated infections. The worldwide expansion of infections induced by colistin-resistant and carbapenemase-producing Enterobacterales (CPE) isolates has been increasingly reported. This study aims to analyze the phenotypic and molecular profiles of 10 colistin-resistant (CR) isolates and 2 pairs of colistin-heteroresistant (ChR) (parental and the corresponding resistant mutants) isolates of *K. pneumoniae* CPE sourced from two hospitals. The phenotypes of strains in the selected collection had been previously characterized. Antimicrobial susceptibility testing was performed using a Vitek 2 Compact system (BioMérieux SA, Marcy l’Etoile, France), the disc diffusion method, and broth microdilution (BMD) for colistin. Whole-genome sequencing (WGS) did not uncover evidence of any mobile colistin resistance (*mcr*) genes, although the *mgrB* gene of seven isolates appeared to be disrupted by insertion sequences (ISKpn25 or ISKpn26). Possible deleterious missense mutations were found in *phoP* (L4F), *phoQ* (Q426L, L26Q, L224Q, Q317K), *pmrB* (R256G, P95L, T157P, V352E), and *crrB* (P151S) genes. The identified isolates belonged to the following clonal lineages: ST101 (n = 6), ST147 (n = 5), ST258 (n = 2), and ST307 (n = 1). All strains harbored IncF plasmids. OXA-48 producers carried IncL and IncR plasmids, while one *bla*_NDM-1_ genome was found to harbor IncC plasmids. Ceftazidime–avibactam remains a therapeutic option for KPC-2 and OXA-48 producers. Resistance to meropenem–vaborbactam has emerged in some *bla*_kPC-2_-carrying isolates. Our study demonstrates that the results of WGS can provide essential evidence for the surveillance of antimicrobial resistance.

## 1. Introduction

Recently, it has been reported that carbapenemase-producing Enterobacterales (CPE) isolates are expanding at an alarming rate around the globe [1,2]. These Gram-negative bacilli are responsible for causing a variety of hospital-acquired and community-onset human infections, with *Klebsiella pneumoniae* being the principal pathogen associated with significant mortality [2,3,4].

Antimicrobial resistance of *K. pneumoniae* strains continues to be problematic [5]. Resistance to carbapenems is frequently associated with resistance to multiple other classes of antibiotics, which leads to limited possibilities for the treatment of infections induced by these multidrug-resistant (MDR) and extensively drug-resistant (XDR) pathogens [3,5,6]. Around the world, the geographical distribution of carbapenem resistance is reported to be heterogeneous [7]. 

The main carbapenem-hydrolyzing enzymes, *K. pneumoniae* carbapenemase (KPC), metallo-β-lactamases (MBL), and oxacillinase-48-like (OXA-48-like), are principally encoded on plasmids and linked with diverse mobile genetic elements [8].

The global propagation of carbapenemases is the result of a continuous interaction between the clonal dispersion of some efficient CPE lineages and the horizontal transfer of their resistance genes [9]. KPC-producing *K. pneumoniae* multilocus sequence type ST258/512 is a recognized “high-risk international clone” [2,10]. In Europe, ST258/512 and three additional clonal lineages, ST11, ST15, and ST101, along with their derivatives, are predominant [2,10], offering stability for carbapenemase genes [9]. The issues that remain unresolved are the distinct success of *K. pneumoniae* ST258 and the special relationship with epidemic resistance plasmids of incompatibility group F (IncF), which have diverse replicon types (FIA, FIB, and FII) [11,12]. The acquired antimicrobial resistance genes located on IncF can rapidly disseminate within species, while those situated on other types of plasmids, such as IncA/C, IncL/M, and IncN, can be transmitted between species [11,12]. The latter three types of plasmids are unusual in *K. pneumoniae* ST258 and have not been determined to contribute to international dominance [10]. IncL/M type plasmids are associated with *bla*_OXA-48-like_ genes, and IncA/C and IncN types with *bla*_NDM_ genes that encode New Delhi metallo-β-lactamases (NDMs) [9,11].

The situation has evolved with the emergence of *K. pneumoniae* strain ST23, which integrates both hypervirulence and resistance to carbapenems, with dramatic clinical consequences [2,4,13]. Biomarkers present on virulence plasmids have been shown to accurately differentiate hypervirulent from classical strains. The presence of the plasmid-associated gene loci *peg-344*, *iro* (salmochelin biosynthesis), *iuc* (aerobactin synthesis), *rmpA* (regulator of mucoid phenotype), and *rmpA2* are highly predictive of strains as hypervirulent [4,13,14]. 

The worldwide escalation of colistin-resistant CPE isolates is of significant concern [15]. Colistin is a reserve antimicrobial agent with bactericidal action, principally as a consequence of interactions with the outer and inner membranes of Gram-negative bacilli [15,16]. Multiple chromosomal mutations constitute the main substrate of acquired resistance to colistin and promote alterations of the outer membrane lipopolysaccharide (LPS) with an abnormal positive charge, which decreases the affinity and action of colistin [17]. This strategy implies that the addition of 4-amino-4-deoxy-L-arabinose (L-Ara4N) and phosphoethanolamine (PEtN) molecules induces the covalent transformation of lipid A of LPS [17]. A panel of various genes modulates these modifications, essentially based on the two-component regulatory systems (TCRSs) *phoPQ* and *pmrAB* and the feedback regulator *mgrB* gene and the colistin resistance regulation (*crrAB*) operon, respectively [17,18]. Transposition of insertion sequences (ISs) such as ISL3 (ISKpn25), IS5 (ISKpn26), ISKpn14, and IS903B from plasmids into the *mgrB* gene mediates the inactivation and disruption of this regulator gene, conferring colistin resistance [19,20,21]. Antibiotic heteroresistance is considered to be the presence of subpopulations with increased resistance within the majority cell population in the same culture [22]. However, a coherent definition and global guidelines for antibiotic heteroresistance detection are lacking, which complicates its diagnosis [22]. Of particular concern is the emergence of transferable colistin resistance via plasmids [5,15,16,18], with the discovery of a series of mobile colistin resistance (*mcr*) genes, *mcr-1* to *mcr-10*, with diverse variants [23] that encode PEtN, which is incorporated into lipid A [15,23].

Despite continuous research activity, there is still an incomplete understanding of the molecular mechanisms of colistin resistance and heteroresistance [16,24], along with limited data regarding their dissemination and impact [5,18,25]. Whole-genome sequencing (WGS) has become an essential modern tool for interrogating both existing and emerging mechanisms of antibiotic resistance, and it has excellent potential in infection control surveillance [26].

Consequently, the aim of the study is to perform a detailed phenotypic and genomic characterization of 10 colistin-resistant (CR) isolates and 2 pairs of colistin-heteroresistant (ChR) (parental and the corresponding resistant mutants) isolates of *K. pneumoniae* CPE, collected from two Romanian hospitals in 2017 and 2021, toward delineating the variety of molecular substrates of colistin resistance and heteroresistance, identifying the genetic determinants of resistance to other classes of antibiotics, confirming the presence of circulating high-risk clones and dominant plasmids, and exploring potential therapeutic options, including novel β-lactam/β-lactamase inhibitor combinations.

## 2. Results

### 2.1. Phenotypic Antimicrobial Susceptibility Testing

All isolates presented resistance to multiple classes of antimicrobial agents, specifically meropenem, ertapenem, doripenem, aztreonam, cephalosporins, old generations of β-lactam/β-lactamase inhibitors, ciprofloxacin, and levofloxacin. The phenotypic results of antibacterial drug testing are given in Table 1.

Of the three aminoglycosides tested, gentamicin was the most active in vitro (n = 8 susceptible isolates), while total resistance was noted for tobramycin (Table 1). Most of the isolates (n = 12) were susceptible to tigecycline and ceftazidime–avibactam (Table 1). Imipenem–relebactam exhibited potent activity against all KPC producers, while meropenem–vaborbactam showed activity against only five out of the eight KPC producers (Table 1).

Minimum inhibitory concentrations (MICs) of colistin obtained by Vitek 2 Compact software are given in Table 1.

Colistin susceptibility test results obtained with other phenotypic methods and reference broth microdilution (BMD) are given in Table 2. MIC values of colistin obtained by BMD ranged from 0.25 to >64 mg/L (Table 2).

### 2.2. Whole-Genome Sequencing Data Analysis

#### 2.2.1. Quality Metrics for the Assembled Draft Genomes

SPAdes assembly metrics for the sequenced genomes are presented in Table 3. They suggest that all the sequences are adequate for downstream analysis.

#### 2.2.2. Genetic Determinants of Antimicrobial Resistance

All genomes belonged to MDR strains, with several defined resistance genes against different categories of antibiotics, along with other antibiotic resistance determinants, such as insertions, point mutations, or substitutions that lead to the disruption of genes, thus leading to resistance to at least six antibiotic classes.

(A) Molecular mechanism of resistance to various antibiotic classes, except colistin

The highest heterogeneity of resistance-conferring elements was noted for β-lactams (n = 13), aminoglycosides (n = 11), and fluoroquinolones (n = 8) (Table 4). Isolates *bla*_NDM_-1-positive BC7_BM and BC1_TM presented the highest numbers of genetic determinants (n = 28 and n = 22, respectively), while the strain BC2_BM OXA-48 producer had the lowest (n = 12) (Table 4).

In agreement with the phenotypic results, the genotypic analysis confirmed the presence of carbapenem-hydrolyzing enzymes responsible for carbapenem resistance.

For cephalosporin resistance, extended-spectrum β-lactamases (ESBLs) were the molecular elements most frequently detected, represented by *bla*_CTX-M-15_ (n = 12), followed by *bla*_SHV-12_ (n = 2). All *bla*_OXA-48_ genes were accompanied by *bla*_CTX-M-15_ genes. The NDM-1 producer isolate BC1_TM co-presented *bla*_CTX-M-15_ and *bla*_SHV-12_ genes. In the genome of isolate BC7_BM, *bla*_NDM-1_, *bla*_CTX-M-15_, and *bla*_CMY-16_ genes were found (Table 4).

Genes encoding the aminoglycoside-modifying enzyme (AME) were the main elements conferring resistance to aminoglycosides in all strains, with the *aac(6′)-Ib* gene being the most commonly encountered (n = 12 strains) (Table 4). All isolates harbored two to six AME genes, and only isolate BC7_BM co-expressed the *rmtC* gene, a type of 16S rRNA methyltransferase (RMT) (Table 4). Amikacin nonsusceptible isolates encoded the following genes: *aac(6′)-Ib* (n = 12), *aac(6′)-Ib-cr5* (n = 2), *aph(3′)-VI* (n = 2), and *rmtC* (n = 1) (Table 4). Gentamicin nonsusceptible strains possessed the *aac(3)-IIe* gene (n = 3), followed by the *aac(3)-IId* gene (n = 1) (Table 4). The *aac(6′)-Ib* gene (n = 6) and the *rmtC* gene (n = 1) were responsible for tobramycin resistance.

The *oqxAB* efflux pump genes and various chromosomal mutations in the quinolone resistance-determining region (QRDR) of *gyrA* and *parC* genes contributed to fluoroquinolone resistance in all the analyzed strains. Six strains were observed with the genotype profile of *gyrA* D87N and *parC* S80I mutations (Table 4). All the isolates carried a *gyrA* S83 mutation, with *gyrA* S83Y identified in six strains and *gyrA* S83I in eight. Additionally, plasmid-mediated quinolone resistance (PMQR) genes were noted: *aac(6′)-Ib-cr5* (n = 2) and *qnrS1* (n = 1) (Table 4).

A number of alterations in the principal outer membrane of porin-encoding genes *ompK36* and *ompK37*, which potentially confer resistance to cephalosporins and carbapenems, identified by ResFinder, are summarized in Appendix A.

Similarly, using BioEdit for sequence alignment, the glycine–aspartic acid duplication (GD) (position 134–135 duplicated in 136–137) in *ompK36* was visualized in five strains (BC2_BM, BC4_BM, BC5_TM, BC9_TM, and BC10_TM). The insertion of ISKpn26 (IS5) upstream of *ompK36* (position -48 versus the start of *ompK36* gene), which leads to *ompK36* inactivation, was visualized in WGS data and further confirmed by Sanger sequencing (data not shown) in the BC3_TM and BC8_BM isolates. The BC1_TM strain possessed an N186K mutation, while BC7_BM carried several single nucleotide polymorphisms (SNPs), along with deletions, insertions, and substitutions. Similarly, the isolates BC3_TM, BC7_BM, and BC8_BM presented SNPs and substitutions in *ompK37*, while an insertion in *ompK35* that produced a frame shift (FS) at amino acid 166 was noted in isolate BC6_BM. ClustalW (BioEdit) alignment of *ompK35*, *ompK36*, and *ompK37* genes is shown in Appendix A.

(B) Molecular determinants of colistin resistance

The genomic results indicate the absence of plasmidial *mcr* genes.

At least one colistin-resistance chromosomal determinant was noted in the *mgrB* gene or in the two-component regulatory systems *phoPQ*, *pmrAB*, and *crrAB* (Table 5).

For the three CR strains (BC1_TM, BC6_BM, and BC9_TM) with the highest colistin MIC value by BMD, 32 mg/L, two distinct classes of insertion sequences, ISL3 (ISKpn25) and IS5 (ISKpn26), were identified to disrupt the *mgrB* gene in three different nucleotide positions, rendering it nonfunctional (Table 5). Four other CR strains had an M27K SNP in the *mgrB* gene. Additionally, other SNPs identified in CR isolates with the potential to change colistin sensitivity were in *phoP* (L4F), *phoQ* (L26Q, Q426L), and *pmrB* genes (P95L, T157P, R256G, V352E) (Table 5).

As in the case of BC6_BM, the two ChR strains and their corresponding resistant mutants harbored ISL3 (ISKpn25) in the same nucleotide position and an R256G SNP in the *pmrB* gene, but with a supplementary presumptive intolerant L224Q SNP in the *phoQ* gene. Notably, only the resistant mutants BC12_TM_B_m and BC14_TM_C_m presented deleterious Q317K SNPs in the *phoQ* gene and P151S in the *crrB* gene (Table 5).

In contrast, no changes to *pmrA* and *crrA* genes resulting in functional amino acid substitutions were identified in any isolates; only changes encoding neutral amino acid substitutions were identified (Table 5).

The Protein Variation Effect Analyzer (PROVEAN) scores for amino acid changes with a potentially harmful influence on the biological function of the analogous proteins are given in Table 6. The sorting intolerant from tolerant (SIFT) algorithm also confirmed the tolerant/intolerant classification.

(C) Association between phenotypic antimicrobial susceptibility test results and genotypic prediction

In the case of strains BC3_TM and BC8_BM, despite being characterized by Vitek 2 Compact as being resistant to gentamicin, no corresponding molecular resistance element was noted (Table 1 and Table 4). Similarly, the instrument indicated that BC1_TM, BC2_BM, BC4_BM, and BC10_TM were resistant to chloramphenicol, but no genetic resistance markers were detected (Table 1 and Table 4). In these two situations, the disc diffusion method indicated susceptibility to the same antimicrobial agents in all cases. Strain BC9_TM was phenotypically susceptible to amikacin and trimethoprim–sulfamethoxazole, but resistance genes *aac(6′)-Ib-cr5* and *dfrA14* were detected. Two isolates, BC7_BM and BC9_TM, were categorized as susceptible to tigecycline, but *tet(A*) and *tet(D)* genes were identified. Two other strains (BC4_BM and BC10_TM) that showed intermediate susceptibility to tigecycline did not carry tetracycline resistance markers. Conversely, isolate BC14_TM_C_m was interpreted as resistant to chloramphenicol based on both testing methods, but no resistance genes were found upon genetic analysis (Table 1 and Table 4). For the remaining antimicrobial compounds, except colistin, no disagreement between phenotypic profiles and molecular elements was noted (Table 1 and Table 4).

Interestingly, in the case of BC3_TM, the colistin susceptibility profile based on Vitek 2 Compact revealed a discrepancy between the CMI results obtained concomitantly by AST-N222 and AST-XN05 cards, though molecular markers of colistin resistance were detected (Table 1 and Table 5). In the case of BC2_BM, disagreement was noted between the Vitek 2 Compact results and the molecular analysis of colistin. All BMD colistin results correlated strongly with genotypic findings, except in the case of two ChR isolates. 

The concordance between molecular and phenotypic susceptibility testing results for Vitek 2 Compact and for disc diffusion was 96.42% (297/308 × 100) and 98.21% (330/336 × 100), respectively.

#### 2.2.3. Molecular Serotyping, Plasmid Replicon Identification, and Sequence Type Determination

Molecular serotyping determined that six strains were KL17;O1v1, five were KL112;O2v2, two were KL106;O2v2, and one isolate belonged to serotype KL102;O2v2 (Table 7). No gene characteristics of hypervirulence were identified by the Virulence Factor Database (VFDB) tool.

The identified replicons covered mainly Col, IncFIA, IncFIB, IncFII, IncC, IncL, IncR, and IncX3 type plasmids, with a total of 12 types (Table 7). Virulence-encoding plasmid replicon types IncFIB(K) and IncFII(K) were identified in 12 strains. More than half of the isolates were positive for ColRNAI. IncFIB(pQil) was observed, along with other replicon types, in six strains (Table 7). The genome of all the OXA-48 producers carried IncL and IncR plasmids, while one *bla*_NDM-1_ genome was associated with the IncC plasmid (Table 7). The correlation between ST258 clones and the IncF-carrying plasmid with FII(K) replicons was ascertained. The IncX3 plasmid was positive in two KPC-2-producing isolates. In the genome of strains BC6_BM, BC11_TM_B_hR, BC12_TM_B_m, BC13_TM_C_hR, and BC14_TM_C_m, two replicons from the same incompatibility class were identified, IncFIB(K) and IncFIB(pQil), suggesting the presence of multi-replicon plasmids along with other replicons. 

In the mapping of raw reads of the two *bla*_NDM-1_ sequenced strains to 2014–2015 Târgu Mureș *bla*_NDM-1_ plasmid sequences, no similarity was found between reference plasmids and *bla*_NDM-1_ strain BC1_TM. However, the historical plasmid pNDM_18ES had 96% coverage with 99.89% identity by the reads of BC7_BM strain, and plasmids 1TM and 6TM had 76% and 73% coverage, respectively, with 99.9% identity.

Four sequence types (STs) were assigned by multilocus sequence typing (MLST): ST101 (n = 6), ST147 (n = 5), ST258 (n = 2), and ST307 (n = 1) (Table 7). The colistin-resistant mutants presented the same ST as those of their parental strains, namely, ST147.

Nine complex types were identified by core genome MLST (cgMLST). Five of them (CT5839, CT5840, CT5853, CT5854, and CT5848) were considered new founders by SeqSphere and were included in the *K. pneumoniae* database of the cgMLST.org Nomenclature Server.

Overall, ST101 isolates presented the highest number of distinct plasmid replicon types (n = 7 types). KL17 and KL112 serotypes were exclusively associated with ST101 and ST147, respectively. 

The minimum spanning tree of the 14 CR and ChR strains revealed three clusters of closely related strains (Figure 1):
MST cluster 1, comprising *bla*_KPC-2_ producers: two pairs of ChR with their corresponding resistant mutants (from Târgu Mureș) and BC6_BM (from Baia Mare), collected in the period 2019–2020.MST cluster 2, consisting of three *bla*_OXA-48_ strains, BC2_BM, BC4_BM, and BC10_TM (two from Baia Mare and one from Târgu Mureș), collected in the period 2018–2019.MST cluster 3, comprising two *bla*_KPC-2_ strains, BC3_TM and BC8_BM (from Târgu Mureș and Baia Mare, respectively), collected in the period 2017–2018.


## 3. Discussion

In this study, we analyzed the phenotypic features, resistome, virulome, and plasmid content of 10 CR and 2 pairs of ChR (parental and the corresponding resistant mutants) isolates of *K*. *pneumoniae* CPE, collected from two Romanian hospitals between 2017 and 2021. This information is of critical epidemiological importance in the context of the continuous global expansion of high-risk MDR *K. pneumoniae* clones [1,2,9,10,27]. Additionally, the European Antimicrobial Resistance Surveillance Network (EARS-Net) received reports of an increasing trend of invasive infections due to *K. pneumoniae* CPE isolates at Romanian hospitals, from 32.3% in 2019 to 48.3% in 2020 [5,28]. The varying degrees of resistance observed between European countries may be partially explained by differences in antimicrobial consumption, with the lowest rates of both resistance and use in Northern and Central Europe [10,28].

To the best of our knowledge, this is the first Romanian study that focuses on evaluating molecular determinants of resistance in CR and ChR *K. pneumoniae* CPE isolates. The results may also contribute to mitigating the spread of antimicrobial resistance in Romanian hospitals.

In line with previous reports [2,3,5], our phenotypic findings reflect the remarkable rates of resistance to several classes of antimicrobial agents; only a few pathogens remain susceptible to gentamicin and trimethoprim–sulfamethoxazole, but most exhibit susceptibility to tigecycline. Unsurprisingly, the two NDM-1-positive isolates were susceptible only to tigecycline. New compounds with targeted action against NDM producers have been developed, but none have yet been approved for clinical use [29]. Additionally, among the new combinations of β-lactam/β-lactamase inhibitors, ceftazidime–avibactam and imipenem/cilastatin–relebactam were found to be reliable treatment options for all of our KPC producers, while ceftazidime–avibactam exhibited supplementary in vitro activity against all *bla*_OXA-48_-carrying isolates. In contrast, the resistance to meropenem–vaborbactam in three out of the eight KPC producers represents an unwelcome occurrence in the context of patients who have not previously been exposed to this promising novel antimicrobial compound. 

The resistome analysis confirms the existence of an extensive repertoire of antibiotic resistance genes, along with insertions or point mutations, and all our isolates possessed determinants of resistance to β-lactams, amikacin, tobramycin, fosfomycin, fluoroquinolones, colistin, and phenicol. Similarly, several previous reports described a great variety of resistance genes in *K. pneumoniae* isolates with the MDR or XDR phenotype [27,30,31,32] and pointed out the potential flexibility of these pathogens to accumulate and exchange antimicrobial resistance [27]. Furthermore, the concurrent carriage of multiple β-lactamase genes with overlapping hydrolytic activity was confirmed in each of our analyzed strains. The implications of this have not yet been entirely elucidated, but it might have provided an evolutionary advantage for *K. pneumoniae* by offering an additional reliable basis for resistance [27]. 

In the current evaluation, concordance of more than 96% between phenotypic and genomic predictions of antimicrobial resistance was observed. Similarly, Ruppe et al., obtained more than 96% concordance in a study that included 187 Enterobacterales strains tested phenotypically using the disc diffusion method [33]. However, despite WGS being considered a robust surveillance tool [26,32], there is still insufficient published evidence to support its use as a highly accurate instrument to predict antimicrobial susceptibility phenotypes from genomic traits, and it is necessary to establish a consensus regarding which database to interrogate for the detection of antimicrobial resistance genes [33,34]. 

Of note, some discrepancies were identified in this study. Two isolates, BC3_TM and BC8_BM, were categorized as resistant to gentamicin by Vitek 2 Compact and susceptible by disc diffusion, but no AME or RMT genes were detected. However, inadequate correspondence between genotypes and inference of the presence of substrates with resistance to aminoglycosides have been extensively reported based on the European Committee on Antimicrobial Susceptibility Testing (EUCAST) clinical breakpoints in association with expert rules [35]. Instead, consistent with Ruppe et al. [33], the BC9_TM strain was found to possess the *aac(6′)-Ib-cr5* gene but without phenotypic expression of amikacin resistance. Vaziri et al. underscored the fundamental role of the *qnrB* gene in the *K. pneumoniae* genome, which, in association with the *aac(6′)-Ib-cr5* gene, contributes to increased resistance to aminoglycosides, fluoroquinolones, and cephalosporins [36], a conclusion that was not supported by our findings. A recent nationwide epidemiological survey conducted in Greece confirmed a divergence between the resistance phenotype and the AME genotype for aminoglycosides, probably as a consequence of various competing resistance mechanisms associated with the distinct catalytic activities of AME genes [37]. In addition, molecular detection of a resistance gene does not necessarily indicate expression and activity [38]. 

The *catA1* acetyltransferase and the nonenzymatic *cmlA5* genes, encoding resistance to chloramphenicol, were detected in most of our strains. However, discrepant results for chloramphenicol were noted in the case of four strains (BC1_TM, BC2_BM, BC4_BM, and BC10_TM), showing resistance based on Vitek 2 Compact and susceptibility based on the Kirby–Bauer method, while no resistance genes were detected by WGS. The contribution of another mechanism responsible for chloramphenicol resistance (high expression of efflux systems) should not be underestimated [32], given that all of our isolates harbor the oqxAB efflux pump. 

Previous studies in Romania have reported on the worrisome emergence of the *rmtC* gene in NDM-1-positive isolates belonging to the order Enterobacterales [39,40]. This observation is supported by the results of the present study, in which isolate BC7_BM was found to co-carry the *rmtC* gene along with *bla*_NDM-1_, *bla*_CTX-M-15_, and *bla*_CMY-16_ resistance genes and express pan-aminoglycoside resistance. Similarly, the co-expression of NDM-1 and *rmtC* genes, along with at least one ESBL-encoding gene in *K. pneumoniae* isolates, has previously been documented in Kenya, Turkey, and Saudi Arabia [41,42,43]. Since the first description of the RMT gene aminoglycoside resistance methylase (*armA*) in *K. pneumoniae* in 2003 [44], eight other plasmid-mediated variants (*rmtA* to *rmtH*) and N1-A1408 methyltransferase (MTase) (*npmA*) have emerged in Gram-negative pathogens in various parts of the world [45]. In contrast to our findings, a study performed in Switzerland between 2017 and 2020, which analyzed 103 carbapenem- and aminoglycoside-resistant Enterobacterales strains, did not find any *K. pneumoniae* isolates harboring the *rmtC* gene; the most frequently identified was the *armA* gene [46]. Among the RMT genes, *armA*, *rmtB*, and *rmtC* genes are distributed worldwide and confer high-level, broad-range aminoglycoside resistance, including against plazomicin, a newly approved aminoglycoside compound that can evade virtually all clinically relevant AMEs, including acetyltransferase (aac), phosphotransferase (aph), and adenylyltransferase (aad or ant) [45]. The association between RMT genes and *bla*_NDM_, *bla*_KPC_, and *mcr* genes is a multifaceted topic because of the possible involvement in the expansive dissemination of extensively pandrug-resistant organisms [45]. 

Among our three *bla*_KPC-2_-carrying isolates resistant to meropenem–vaborbactam (BC3_TM, BC5_TM, and BC8_BM), detailed molecular analysis verified their affiliation with the international high-risk lineages ST258 and ST101, with evidence of either GD134-135 duplication in the *ompK36* gene or insertion of ISKpn26 (IS5) upstream of *ompK36* in association with SNPs and substitutions in the *ompK37* porin gene. Furthermore, even though BC6_BM showed susceptibility to meropenem–vaborbactam, an FS mutation at amino acid 166 in *ompK35* was recorded. These mutations are responsible for alterations in permeability that contribute to reduced susceptibility to meropenem–vaborbactam, as previously reported [47,48,49]. Meropenem–vaborbactam penetrates the outer membrane of *K. pneumoniae* mainly through ompK35 and ompK36 porins, but vaborbactam prefers the latter, which has a narrower inner channel [50]. However, the single presence of a nonfunctional ompK35 porin, without mutation in ompK36, is associated with MIC values of ≤0.06 mg/L meropenem–vaborbactam [47], and a higher degree of meropenem–vaborbactam inactivation was demonstrated only upon ompK36 porin loss of function, either in isolation or concurrently with ompK35 [50]. 

In agreement with other reports from diverse geographical regions [30,31,32], chromosomal mutations associated with colistin resistance were revealed in all examined strains in the current study, and no plasmid-borne *mcr* genes were identified. 

MgrB, a small negative regulatory transmembrane protein, represses *phoPQ* signaling, but the expression of this TCRS regulator is upregulated in the presence of a modified *mgrB* gene, and it remodels LPS through the addition of cationic L-Ara4N, which prevents colistin molecules attaching to the LPS membrane [17,18,51]. Mutation of *mgrB* gene was the basis for resistance most frequently encountered in our study (11 out of the total strains) and was induced by either ISKpn25 (n = 6), ISKpn26 (n = 1), or a missense mutation M27K (n = 4), predicted as deleterious by bioinformatics tools. Several recent studies have highlighted the pivotal role of ISs in contributing to the emergence of colistin resistance by disrupting the *mgrB* gene [19,20,21,30,31,32]. Our findings provide evidence that plasmid IncFIB(pQil) encodes ISkpn25, while the ISKpn26 element is associated with IncFIA(HI1) and IncR plasmids. Fordham et al. demonstrated similar connections between these IS elements and their companion plasmid families, with the exception of the relationship between ISKpn26 and IncFII(pHN7A8) plasmids [19]. The M27K mutation in the *mgrB* gene was also reported to mediate colistin resistance in a previous study [52]. In our analysis, only strain BC5_TM presented the M27K mutation as the sole potential marker of resistance to colistin. In contrast, Liu et al. concluded that the expression of MgrB protein was not affected by the M27K mutation, even though the strain exhibited an MIC of 32 mg/L colistin [53]. Inactivation of *mgrB* has also been shown to promote virulence in *K. pneumoniae* isolates by suppressing the initiation of host defense reactions and limiting the action of multiple antimicrobial peptides [54]. 

In comparison with prior studies, our analysis indicates the occurrence of five previously unreported point mutations in *phoP* and *phoQ* genes, which potentially mediate colistin resistance and heteroresistance. We observed an alteration in the *phoP* gene induced by a novel deleterious L4F substitution in the BC2_BM isolate, while several new intolerant SNPs (L26Q, L224Q, Q426L, and Q317K), as anticipated by bioinformatics tools, were detected in *phoQ* in both CR and ChR strains. Meanwhile, other researchers have mentioned the deleterious L26Q substitution in the *phoP* gene [55]. Furthermore, some mutations closely related to those above were detected in the *phoQ* gene (such as L30Q, L26P, L96R, and L257P substitutions) [31,56,57,58] and the *phoP* gene (V3F) [58] and have previously been described as mediating colistin resistance.

Within the complex chromosomal cascade mechanism that confers colistin resistance, *phoP* can also stimulate the production of PmrA protein, which belongs to the *pmrAB* TCRS, either directly or indirectly through the adaptor PmrD protein, leading to the addition of cationic pEtN and L-Ara4N moieties to LPS [18,51,59]. In our investigation, only *pmrB*, as part of the *pmrAB* TCRS, was subject to diverse mutations (R256G, T157P, P95L, and the newly reported V352E). The R256G substitution has been widely reported to contribute to colistin resistance [30,31,32,53], but this deleterious SNP has also been found in polymyxin-susceptible strains [30,58], suggesting that this alteration alone might not be sufficient to increase MIC values for colistin [58]. Interestingly, all seven isolates detected with this substitution presented at least an additional colistin-resistant element and phenotypically expressed resistance, except the two parental ChR strains, with MIC ≤0.5 mg/L colistin by BMD. In addition, T157P substitution has been reported to disrupt the α-helix secondary structure of mutated PmrB protein, with consecutive activation of PmrA [60], and this SNP was mentioned by several authors [32,52,56,60]. In agreement with the analysis of KPC-3-producing Colombia strains by Jajol et al. [60], our KPC-2-producing BC8_BM strain, with the same T157P mutation, belongs to the same ST258 clone. The two PmrB mutations, R256G and T157P, are among the most frequently reported as being associated with colistin resistance [61], and the rare P95L alteration in PmrB was confirmed to confer resistance by complementation assay [62].

Notably, our BC2_BM and BC3_TM isolates, with colistin-susceptible results by Vitek 2 Compact AST N222 cards but an MIC value of ≥8 mg/L colistin by BMD, did not present any *mgrB* gene disruption by ISs. Consequently, in the first case, a combination of mutations in *mgrB* (M27K), *phoP* (L4F), and *phoQ* (Q426L) genes was noted, while in the second case, an L26Q substitution in the *phoQ* gene was associated with R256G mutation in the *pmrB* gene. However, several reports have mentioned that Vitek 2 Compact is inappropriate for colistin susceptibility testing [63,64,65], especially for strains showing an MIC > 1 mg/L by BMD [63]. The manufacturer of Vitek 2 Compact has also recommended using an alternative method prior to reporting colistin results obtained with several types of testing cards in this automated system [66], at least until this issue can be addressed through technology upgrades to be implemented in the future.

Moreover, the *crrAB* TCRS can activate Pmr A [51,59], and gain-of-function mutations in *crrB* alone can activate the expression of genes leading to colistin resistance without any contribution from the *pmrAB* TCRS [59]. Interestingly, our findings revealed a rare P151S substitution in the *crrB* gene only in the resistant mutant BC14_TM_C_m isolate, which, in contrast to its susceptible parental strain BC13_TM_C_hR, expressed an MIC value of >64 mg/L colistin. The P151S substitution detected in the putative histidine kinase domain has been validated to induce elevated resistance to colistin [67]. Similarly, Jajol et al. reported a mutation with a subtle difference, P151L in *crrB*, as conferring colistin resistance [52], while Pitt et al. confirmed a P158R substitution in the same gene [30].

Remarkably, another significant aspect of our study is the molecular basis of the two pairs of ChR (parental and the corresponding resistant mutants) isolates, which are reported for the first time in Romania [65]. These two pairs of strains co-harbored an *mgrB* alteration caused by an ISKpn25 element insertion at the same position (with the deletion of nucleotides 1–5) and an R256G substitution in *pmrB*. An identical molecular profile was observed with the BC6_BM strain, which, unlike the two parental ChR strains, had an MIC value of 32 mg/L colistin indicated by BMD. In particular, the two pairs of isolates were supplementarily accompanied by a new L224Q substitution in *phoQ*, and the two mutants presented either an additional amino acid change P151S in *crrB* or Q317K in *phoQ*. Despite the presence of a disrupted *mgrB* gene in the two parental ChR strains, a possible explanation for the results of colistin susceptibility according to BMD might be the potential suppressor effect of the L224Q mutation in *phoQ*. In addition, BMD has been considered an unreliable method for the detection of colistin heteroresistance [25,68,69]. Pitt el al., reported that *K. pneumoniae* isolates exclusively carrying an ISKpn26-like element exhibit an MIC of ≥64 mg/L colistin, while the introduction of a mutated *phoP* (P47L or A95S) or *phoQ* (N253T or V446G) gene into an IS *mgrB*-altered strain led to a decrease in the MIC [30]. These mutations have been shown to disturb the pathways involved in *phoQ*, *phoP*, and *pmrD* expression [30]. However, the complex and challenging mechanisms underlying colistin resistance have not been entirely decoded, and accurate detection accompanied by functional analysis is required in order to clarify their influence on resistance [17,30,51,59]. Moreover, there are still insufficient data on the genetic basis of colistin heteroresistance, although there is evidence that mutations in *phoPQ*, *pmrAB*, and *mgrB* regulatory systems or in *lpxM* and *yciM* alleles are involved [61,69,70,71,72] and that the *mcr-1* gene is not connected with this phenomenon based on PCR results [68,71].

The strains included in our collection were concentrated in either established (ST258, ST101, and ST147) or emerging (ST307) international clonal lineages [11,27,73,74]. The circulation of bacterial clones ST258, ST101, and ST307 in Romania has been previously documented in both clinical and wastewater specimens [39,40,75]. The present analysis reaffirms the prominent relationship between ST258 and *bla*_KPC_ genes [27], whereas our strains assigned to ST101 carried *bla*_OXA-48_, *bla*_KPC-2_, and *bla*_NDM-1_ genes, consistent with Palmieri et al. [76]. *K. pneumoniae* genotype ST101 has been associated with resistance to carbapenem, colistin, aminoglycosides, fluoroquinolones, and fosfomycin, with the potential to become a “perfect storm” clone, considering the similarity with the genetic profiles of hypervirulent strains [77]. In addition, a recent Romanian study demonstrated that it is able to persist in wastewater samples after chlorine treatment [78]. All of our ST101 isolates expressed capsular KL17 and somatic O1v1 antigens, and these findings are also supported by other studies [77,79]. *K. pneumoniae* ST147 and ST307 have been associated with pandrug resistance and have been reported from endemic regions as well as in global nosocomial outbreaks, with proven links to KPC, NDM, OXA-48-like, and Verona integron-encoded MBL (VIM) [73]. In agreement with Peirano et al. [73], our six strains, assigned to ST147 and ST307, were found to carry the ESBL *bla*_CTX-M-15_ gene along with *gyrA* S83I and *parC* S80I mutations. Contrary to a recent report on an outbreak in northeast Germany induced by *K. pneumoniae* ST307 co-harboring *bla*_NDM-1_ and *bla*_OXA-48_ genes and colistin resistance [74], the *bla*_OXA-48_ gene was not detected in our ST307 isolate, nor were any virulence genes or elements conferring susceptibility to chloramphenicol detected.

It is noteworthy that the two parental strains, BC11_TM_B_hR and BC13_TM_C_hR, showed the same phenotypic and genomic profiles belonged to the successful ST147 clone, carried the same *mgrB*, *phoQ*, *pmrB*, and *crrB* alterations, and were recovered from the same intensive care unit (ICU) approximately 3 weeks apart from different patients, all of which are indicative of silent clonal expansion and intraward dissemination. There is also concern regarding the close relationship of these two ChR isolates from Târgu Mureș with the BC6_BM strain isolated in Baia Mare, suggesting the interregional propagation of ST147. Furthermore, the OXA-48-producer cluster ST101 revealed both interward and interregional spread, while the KPC-2-producer cluster ST258 showed interregional transmission. The constituent isolates of each cluster possessed similar or almost similar plasmids, implying plasmid-mediated propagation of *bla*_OXA-48_ and *bla*_KPC-2_ genes.

The interdependence between KPC-2-producing isolates with FII(K) and IncX3 plasmids and between OXA-48 producers with IncL and IncR plasmids concurs with previous observations made by Becker et al. in Germany [80]. In contrast to a prior study conducted at the same medical institution in Târgu Mureș on strains collected between 2012 and 2013 that demonstrated the presence of the IncR plasmid replicon in five *K. pneumoniae* NDM-1-positive isolates [81], our two NDM-1 producers did not harbor this replicon type. Moreover, a significant match between the BC7_BM strain and the reference plasmids suggests the persistence and evolution of a *bla*_NDM_ plasmid in this geographical area, which was previously named pKOX_NDM1-like by Phan et al. [82]. Additionally, the concurrent carriage of IncFIB(K) and IncFIB(pQil) replicons belonging to the same incompatibility class is in agreement with the results of Villa et al., demonstrating the great versatility of IncF plasmids [83]. These extrachromosomal DNA molecules often exhibit a multi-replicon status [12,83], which enables the acquisition of plasmids harboring incompatible replicons when replication is promoted by a compatible replicon [83]. However, the second-generation raw reads used in our research did not offer the same possibility to construct the entire sequence of a plasmid as in the case of long-read sequencing [32,38].

The present investigation has some limitations. The retrospective nature of the study did not allow us to provide a reliable picture of all significant clinical and therapeutic aspects or previous hospitalizations, which was beyond the scope of this manuscript. However, the data on previous colistin therapy in patients diagnosed with ChR strains may be found elsewhere [65]. BMD was performed only for colistin as the research was focused on detecting the molecular basis of resistance and heteroresistance to this antimicrobial agent. Complementation experiments with wild-type alleles to validate novel mutations potentially conferring colistin resistance and heteroresistance were not performed because of logistical constraints and should be included in future investigations.

### Future Directions

Future studies should be oriented toward the reliable detection of both existing and novel mutations, combined with functional analysis, to establish their real contribution to colistin resistance and heteroresistance. Additional research should be conducted to expand our knowledge of the complex mechanisms underlying colistin resistance and heteroresistance; for the last phenomenon to establish clinical relevance, we should formulate a harmonized international definition and develop a standardized detection methodology. Furthermore, investigations using long-read sequencing technology will offer an adequate resolution for exploring plasmid structures.

## 4. Materials and Methods

### 4.1. Bacterial Strains, Setting, and Design of the Study

A total of 10 unique clinical CR *K. pneumoniae* CPE strains were analyzed: 5 selected from patients admitted to Dr. Constantin Opriș County Emergency Hospital, Baia Mare, Romania, between January 2017 and April 2021 and 5 from patients at Târgu Mureș County Emergency Clinical Hospital, Romania, between January 2017 and April 2019. Additionally, 2 ChR *K. pneumoniae* strains obtained from Târgu Mureș in March 2019 and their corresponding colistin-resistant mutants were included in the study. The collection comprising the 10 CR and 2 ChR isolates has been phenotypically characterized in an earlier study [65]. 

The first medical center is a public 920-bed general acute care nonteaching hospital in the northwest region of Romania, and the second is a large 1089-bed teaching hospital located in Transylvania, in the central region; they are located approximately 200 km apart.

CR *K. pneumoniae* CPE isolates were randomly selected based on (i) MIC values greater than 2 mg/L colistin (categorized as resistance) [84], as determined using the BMD method; (ii) diverse specimens collected from various hospital wards; (iii) carbapenemase types identified phenotypically (a set of n = 2 KPC, n = 2 OXA-48-like, and n = 1 MBL for each medical institution); and (iv) date of collection.

Pathogens were mostly isolated from patients in the ICU (n = 6) and were obtained from different anatomical sites, as summarized in Table 8.

### 4.2. Demographic Data of Patients

Data from electronic medical records available in the 2 laboratories are presented in Table 8.

### 4.3. Phenotypic Bacterial Identification and Antimicrobial Susceptibility Testing

All strains were identified at the species level using standard techniques and a Vitek 2 Compact system (BioMérieux SA, Marcy l’Etoile France). AST-XN05 and AST-N233 pair testing cards, starting from the same inoculum, were used with the Vitek 2 Compact system, and AST-N222 cards were added for particular strains. Vitek 2 Compact version 9.02 software was used. The Kirby–Bauer disc diffusion method was used in all cases. Meropenem–vaborbactam (30 µg), imipenem–relebactam (35 µg), ceftazidime–avibactam (14 µg), and doripenem (10 µg) were tested exclusively by disc diffusion.

All *K. pneumoniae* CPE strains were additionally assessed for colistin resistance by BMD and using the following 5 phenotypic methods, as described elsewhere [65]: Micronaut MIC-Strip (Merlin Diagnostika GmbH, Bornheim-Hersel, Germany), Etest gradient diffusion strip on Mueller Hinton E agar (MHE) (BioMérieux SA, Marcy l’Etoile, France), ChromID Colistin R agar (COLR) assay (BioMérieux SA, Marcy l’Etoile, France), Rapid Polymyxin NP test (ELITechGroup, Signes, France), and colistin broth disc elution. Moreover, the 2 ChR strains were further confirmed using the population analysis profiling (PAP) assay, as also described in [65].

The EUCAST breakpoint [84] was applied for the interpretation of all antibiotic susceptibility test results, except for tigecycline, for which US Food and Drug Administration (FDA) criteria were adopted [85]. 

The modified carbapenem inactivation method (mCIM) [86,87] and the combination disc test (KPC, MBL, and OXA-48 Confirm Kit, Rosco Diagnostica, Denmark) were applied to phenotypically categorize the carbapenemase producers, as described elsewhere [65].

Concordance between phenotypic and genotypic susceptibility test results was calculated individually for Vitek 2 Compact (14 isolates × 22 antimicrobial agents = 308 combinations) and the disc diffusion method (14 isolates × 24 agents = 336 combinations) as the proportion of concordant results in the total combinations tested by each method. In this study, results of susceptibility or susceptibility to increased exposure were considered to be discrepant if a resistance molecular marker was noted or resistance results were not linked with genetic elements.

Routine and extended quality control were performed according to EUCAST [84] with the following reference strains: *Escherichia coli* ATCC 25922, *E. coli* NCTC 13846, *E. coli* ATCC 35218, *K. pneumoniae* ATCC 700603, and *K. pneumoniae* BAA 2814. Other details regarding quality control for all methods used can be found elsewhere [65].

The isolates were frozen at −70 °C and subcultured twice on solid medium before additional testing.

### 4.4. Genotypic Characterization Using WGS

#### 4.4.1. Sequencing and Assembly of Draft Genomes

The 14 genomes of CR, ChR, and corresponding resistant mutants of CPE strains were sequenced using the Ion Torrent PGM platform (Thermo Fisher Scientific, Waltham, MA, USA) according to the 400 bp protocol for library preparation, which includes enzymatic shearing, Ion OneTouch2 emulsion PCR, enrichment, and Hi-Q View sequencing kits (Thermo Fisher Scientific). The sequences obtained were subjected to de novo assembly into contigs using the Assembler SPAdes plugin version 5.12 (21, 33, 55, 77, and 99 k-mers) [88] installed on the Ion Torrent Server. For epidemiological interconnections and worldwide distribution, all the raw reads in the study were deposited in the publicly available European Nucleotide Archive (ENA) database under project PRJEB53146 (ERR9860230–ERR9860243) (https://www.ebi.ac.uk/ena/browser/home, accessed on 11 August 2022). 

WGS data were analyzed using both commercial and free online bioinformatics tools.

#### 4.4.2. Resistome Analysis

For antibiotic resistance characterization, we used AMRFinderPlus version 1.1 [89] through Ridom SeqSphere+ commercial software (Ridom GmbH). The analysis of antimicrobial resistance determinants was complemented with ResFinder version 4.1 [90,91], which is available on the public Center of Genomic Epidemiology (CGE) server (http://www.genomicepidemiology.org/services/, accessed on 20 April 2022) (identity 85%, minimum length 60%), the Basic Local Alignment Search Tool (BLAST) search against reference sequences from the National Center for Biotechnology Information (NCBI) nucleotide repository, and BioEdit software version 7.0.5.3. [92].

Plasmid-mediated *mcr* genes, the presence and integrity of TCRS genes (*phoPQ, pmrAB,* and *crrAB*), and the regulatory transmembrane protein-coding *mgrB* gene were investigated to identify colistin resistance determinants. ISfinder was used to depict the insertion sequence (IS) types that led to gene disruption [93]. In order to predict whether amino acid substitutions identified in PhoPQ, PmrAB, and CcrAB had an impact on the biological function of the proteins, 2 bioinformatics tools were applied: the Protein Variation Effect Analyzer (PROVEAN) version 1.1 [94], with a default score threshold set at -2.5 for binary classification, and the sorting intolerant from tolerant (SIFT) algorithm [95], with default parameters. The corresponding genes from colistin-susceptible *K. pneumoniae* NC_009648.1 were used as references.

#### 4.4.3. Molecular Typing of Isolates and Plasmid Profiling

In order to depict the allelic profile of the genomes, MLST, ST [96], and cgMLST complex type (CT) genes (2358 target genes) were used with Ridom SeqSphere+ software [97]. For cgMLST, a default threshold of 15 allele differences was selected, and a minimum spanning tree with the option pairwise ignoring missing values was generated. 

Capsular serotyping was conducted using the Kaptive web interface (https://kaptive-web.erc.monash.edu/, accessed on 16 June 2022) [98]. The Virulence Factor Database (VFDB) [14], included in SeqSphere+, was interrogated for the detection of genes characteristic of hypervirulent strains.

Plasmid replicons were detected using the web-based PlasmidFinder version 2.1 [99] on the CGE server (with default parameters). In an article published in 2018, Phan et al. identified *bla*_NDM_ plasmids circulating during the period 2014–2015 in the town of Târgu Mureș [82]. Since these plasmid sequences are publicly available, Burrows–Wheeler alignment (BWA) was adopted for the mapping of raw reads from the 2 *bla*_NDM-1_-positive strains (BC1_TM and BC7_BM) to 3 reference plasmid sequences from 2014–2015 for comparison, namely, *K*. *pneumoniae* strain 1TM plasmid pNDM_1TM (MF042353.1), *K. pneumoniae* strain 6TM plasmid pNDM_6TM (MF042354.1), and *K. pneumoniae* strain 18ES plasmid pNDM_18ES (MF042350.1).

## 5. Conclusions

This study highlights the significant challenges involved in the phenotypic and molecular diagnoses of colistin resistance and heteroresistance. Each of our isolates presented at least one mutation in the *mgrB, phoPQ, pmrAB*, or *crrAB* gene, predicted to confer colistin resistance. We report on a novel and rare potential suppressor mutation in *phoQ*, L224Q, with possible involvement in heteroresistance to colistin, which in the presence of an altered *mgrB*, modified by IS, leads to low MIC values for colistin, as determined using BMD and other phenotypic methods aside from the Rapid Polymyxin NP test and PAP assays. Evidence for the silent intrahospital dissemination of heteroresistant mutant ST147 clones is thus provided. 

This research underlines the importance of WGS in identifying a catalog of molecular markers of various classes of antibiotics coupled with descriptions of circulating plasmids and clones, data that can be useful when investigating the dynamics of the dispersion of *K. pneumoniae* MDR isolates, in addition to being an innovative and promising strategy for continuous surveillance of antimicrobial resistance in order to limit worldwide transmission. 

Ceftazidime-avibactam, imipenem/cilastatin-relebactam, meropenem-vaborbactam, tigecycline, gentamicin, and trimethoprim-sulfamethoxazole are still potential in vitro agents with activity against some of the studied pathogens.

## Figures and Tables

**Figure 1 antibiotics-11-01171-f001:**
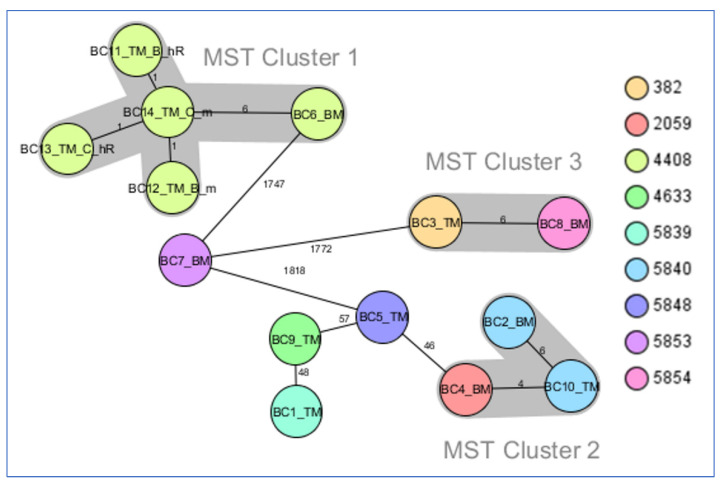
Minimum spanning tree for *K. pneumoniae* strains investigated in this study generated from cgMLST data. Ridom SeqSphere+ MST for 14 samples based on 2358 columns, pairwise ignoring missing values, logarithmic scale. Distance based on columns from *K. pneumoniae* sensu lato cgMLST (2358). MST cluster distance threshold: 15. Nodes colored by complex type.

**Table 1 antibiotics-11-01171-t001:** Antimicrobial resistance profiles obtained by Vitek 2 Compact software and the disc diffusion method * for all strains.

		Bacterial Strain													
Antimicrobial Agent		BC1_TM	BC2_BM	BC3_TM	BC4_BM	BC5_TM	BC6_BM	BC7_BM	BC8_BM	BC9_TM	BC10_TM	BC11_TM_B_hR	BC12_TM_B_m	BC13_TM_C_hR	BC14_TM_C_m
** Ampicillin **		**R**	**R**	**R**	**R**	**R**	**R**	**R**	**R**	**R**	**R**	**R**	**R**	**R**	**R**
**Amoxicillin–clavulanic acid**		**R**	**R**	**R**	**R**	**R**	**R**	**R**	**R**	**R**	**R**	**R**	**R**	**R**	**R**
**Piperacillin–tazobactam**		**R**	**R**	**R**	**R**	**R**	**R**	**R**	**R**	**R**	**R**	**R**	**R**	**R**	**R**
**Ceftazidime–avibactam ***		**R**	**S**	**S**	**S**	**S**	**S**	**R**	**S**	**S**	**S**	**S**	**S**	**S**	**S**
**Meropenem–vaborbactam ***		**R**	**R**	**R**	**R**	**R**	**S**	**R**	**R**	**R**	**R**	**S**	**S**	**S**	**S**
**Imipenem/cilastatin–relebactam ***		**R**	**I**	**S**	**R**	**S**	**S**	**R**	**S**	**R**	**R**	**S**	**S**	**S**	**S**
** Cefuroxime **		**R**	**R**	**R**	**R**	**R**	**R**	**R**	**R**	**R**	**R**	**R**	**R**	**R**	**R**
** Cefoxitin **		**R**	**R**	**R**	**R**	**R**	**R**	**R**	**R**	**R**	**R**	**R**	**R**	**R**	**R**
** Cefotaxime **		**R**	**R**	**R**	**R**	**R**	**R**	**R**	**R**	**R**	**R**	**R**	**R**	**R**	**R**
** Ceftazidime **		**R**	**R**	**R**	**R**	**R**	**R**	**R**	**R**	**R**	**R**	**R**	**R**	**R**	**R**
** Ceftriaxone **		**R**	**R**	**R**	**R**	**R**	**R**	**R**	**R**	**R**	**R**	**R**	**R**	**R**	**R**
** Cefepime **		**R**	**R**	**R**	**R**	**R**	**R**	**R**	**R**	**R**	**R**	**R**	**R**	**R**	**R**
**Aztreonam**		**R**	**R**	**R**	**R**	**R**	**R**	**R**	**R**	**R**	**R**	**R**	**R**	**R**	**R**
** Meropenem **		**R**	**R**	**R**	**R**	**R**	**R**	**R**	**R**	**R**	**R**	**R**	**R**	**R**	**R**
** Imipenem **		**R**	**S■**	**R**	**I▲**	**R**	**R**	**R**	**R**	**R**	**S■**	**R**	**R**	**R**	**R**
** Ertapenem **		**R**	**R**	**R**	**R**	**R**	**R**	**R**	**R**	**R**	**R**	**R**	**R**	**R**	**R**
** Doripenem * **		**R**	**R**	**R**	**R**	**R**	**R**	**R**	**R**	**R**	**R**	**R**	**R**	**R**	**R**
**Gentamicin**		**R**	**S**	**R●**	**R**	**S**	**S**	**R**	**R●**	**R**	**S**	**S**	**S**	**S**	**S**
**Amikacin**		**R**	**R**	**R**	**R**	**R**	**R**	**R**	**R**	**S**	**R**	**R**	**R**	**R**	**R**
**Tobramycin**		**R**	**R**	**R**	**R**	**R**	**R**	**R**	**R**	**R**	**R**	**R**	**R**	**R**	**R**
** Ciprofloxacin **		**R**	**R**	**R**	**R**	**R**	**R**	**R**	**R**	**R**	**R**	**R**	**R**	**R**	**R**
** Levofloxacin **		**R**	**R**	**R**	**R**	**R**	**R**	**R**	**R**	**R**	**R**	**R**	**R**	**R**	**R**
**Trimethoprim–sulfamethoxazole**		**R**	**S**	**R**	**S**	**R**	**R**	**R**	**R**	**S**	**S**	**R**	**R**	**R**	**R**
** Chloramphenicol **		**R●**	**R●**	**R**	**R●**	**R**	**R**	**R**	**R**	**S**	**R●**	**R**	**R**	**R**	**R**
**Tigecycline ****		**S**	**S**	**S**	**I**	**S**	**S**	**S**	**S**	**S**	**I**	**S**	**S**	**S**	**S**
** Colistin ** ** ** (MIC mg/L) **	**AST-N222**	**NT**	**≤0.5 S**	**2** **S**	**NT**	**NT**	**NT**	**NT**	**NT**	**NT**	**NT**	**≤0.5 S**	**≥16 R**	**≤0.5 S**	**≥16 R**
	**AST-XN05**	**≥16** **R**	**1** **S**	**4** **R**	**≥1** **6 R**	**8** **R**	**≥16** **R**	**≥16 R**	**≥16 R**	**≥16 R**	**≥16 R**	**≤0.5 S**	**≥16 R**	**≤0.5 S**	**≥16 R**

**Notes:** * Only disc diffusion method; ** only Vitek 2 Compact version 9.02 software. R (red): resistance; I (yellow): susceptible to increased exposure (for all drugs tested, except tigecycline) or intermediate susceptibility (only for tigecycline); S (green): susceptible; NT (pink): not tested. ● Discrepancy between Vitek 2 Compact, indicating resistance, and disc diffusion, categorizing strain as susceptible. ■ Discrepancy between Vitek 2 Compact, indicating susceptibility, and disc diffusion, categorizing strain BC2_BM as susceptible to increased exposure and isolate BC10_TM as resistant. ▲ Discrepancy between Vitek 2 Compact, indicating susceptibility to increased exposure, and disc diffusion, categorizing strain BC4_BM as resistant. MIC, minimum inhibitory concentration; hR, heteroresistant; m, mutant. The font color of antimicrobial agents belonging to distinct groups of beta-lactams or different classes of antibiotics is marked alternatively in blue and black color.

**Table 2 antibiotics-11-01171-t002:** Colistin testing results obtained using different assays.

**Bacterial Isolate**	**Micronaut** **MIC-Strip** **(MIC mg/L)**	**Etest, MHE** **(MIC mg/L)**	**COLR** **Medium**	**Rapid** **Polymyxin** **NP Test**	**CBDE** **(MIC mg/L)**	**BMD** **(MIC mg/L)**
BC1_TM	32	4	Positive	Positive	≥4	32
BC2_BM	16	2	Positive	Positive	≥4	8
BC3_TM	8	2	Positive	Positive	≥4	16
BC4_BM	16	4	Positive	Positive	≥4	16
BC5_TM	8	4	Positive	Positive	≥4	8
BC6_BM	32	16	Positive	Positive	≥4	32
BC7_BM	16	8	Positive	Positive	≥4	16
BC8_BM	8	2	Positive	Positive	≥4	8
BC9_TM	32	32	Positive	Positive	≥4	32
BC10_TM	16	4	Positive	Positive	≥4	16
BC11_TM_B_hR	0.5	0.25	Negative	Positive	≤1	0.25
BC12_TM_B_m	>64	192	Positive	Positive	≥4	>64
BC13_TM_C_hR	0.25	0.25	Negative	Positive	≤1	0.5
BC14_TM_C_m	>64	128	Positive	Positive	≥4	>64

MIC, minimum inhibitory concentration; MHE, Mueller Hinton E agar; COLR, ChromID Colistin R agar; CBDE, colistin broth disc elution; BMD, reference broth microdilution; hR, heteroresistant; m, mutant.

**Table 3 antibiotics-11-01171-t003:** Quality control parameters for draft genomes.

Bacterial Strain ID	Number of Contigs ≥ 200 bp	N50	Draft Genome Length (bp)
BC1_TM	122	173,972	5,474,448
BC2_BM	151	174,945	5,948,608
BC3_TM	177	203,739	5,596,147
BC4_BM	154	180,250	5,878,102
BC5_TM	157	171,932	5,749,801
BC6_BM	135	204,587	5,682,036
BC7_BM	124	197,811	5,731,764
BC8_BM	131	172,882	5,655,780
BC9_TM	141	173,978	5,589,246
BC10_TM	159	204,669	5,905,163
BC11_TM_B_hR	183	170,134	5,637,259
BC12_TM_B_m	215	172,730	5,669,127
BC13_TM_C_hR	188	151,322	5,628,400
BC14_TM_C_m	166	206,499	5,692,480

hR, heteroresistant; m, mutant.

**Table 4 antibiotics-11-01171-t004:** Antimicrobial resistance determinants (except colistin) detected in all sequenced genomes.

		Bacterial Strain													
Class	Gene	BC1_TM	BC2_BM	BC3_TM	BC4_BM	BC5_TM	BC6_BM	BC7_BM	BC8_BM	BC9_TM	BC10_TM	BC11_TM_B_hR	BC12_TM_B_m	BC13_TM_C_hR	BC14_TM_C_m
** β-L **	** * bla * ** ** _ CMY-16 _ **							** x **							
	** * bla * ** ** _ CTX-M-15 _ **	** x **	** x **		** x **	** x **	** x **	** x **		** x **	** x **	** x **	** x **	** x **	** x **
	** * bla * ** ** _ KPC-2 _ **			** x **		** x **	** x **		** x **			** x **	** x **	** x **	** x **
	** * bla * ** ** _ NDM-1 _ **	** x **						** x **							
	** * bla * ** ** _ OXA-1 _ **							** x **		** x **					
	** * bla * ** ** _ OXA-9 _ **		** x **		** x **						** x **	** x **	** x **	** x **	** x **
	** * bla * ** ** _ OXA-10 _ **							** x **							
	** * bla * ** ** _ OXA-48 _ **		** x **		** x **					** x **	** x **				
	** * bla * ** ** _ SHV-1 _ **		** x **		** x **	** x **				** x **	** x **				
	** * bla * ** ** _ SHV-11 _ **						** x **		** x **			** x **	** x **	** x **	** x **
	** * bla * ** ** _ SHV-12 _ **	** x **		** x **											
	** * bla * ** ** _ SHV-28 _ **							** x **							
	** * bla * ** ** _ TEM-1 _ **	** x **		** x **	** x **	** x **	** x **	** x **		** x **	** x **	** x **	** x **	** x **	** x **
**Ag**	** *aac(3)-IId* **	**x^G^**													
	** *aac(3)-IIe* **				**x^G^**			**x^G^**		**x^G^**					
	** *aac(6′)-Ib* **	**x^A,T,K^**	**x^A,T,K^**	**x^A,T,K^**	**x^A,T,K^**	**x^A,T,K^**	**x^A,T,K^**		**x^A,T,K^**		**x^A,T,K^**	**x^A,T,K^**	**x^A,T,K^**	**x^A,T,K^**	**x^A,T,K^**
	** *aac(6′)-Ib-cr5* **							**x^A,T,K^**		**x^A,T,K^**					
	** *aadA1* **	**x^S^**	**x^S^**		**x^S^**		**x^S^**	**x^S^**			**x^S^**				
	** *aadA2* **			**x^S^**		**x^S^**	**x^S^**		**x^S^**			**x^S^**	**x^S^**	**x^S^**	**x^S^**
	** *aph(3′)-Ia* **			**x^K^**		**x^K^**	**x^K^**		**x^K^**			**x^K^**	**x^K^**	**x^K^**	**x^K^**
	** *aph(3′)-VI* **	**x^A,K^**						**x^A,K^**							
	** *aph(3′’)-Ib* **	**x^S^**						**x^S^**							
	** *aph(6)-Id* **	**x^S^**													
	** *rmtC* **							**x^A,T,H,K,KS,S,SP^**							
** Gy **	** * ble * **	** x **						** x **							
**Fos**	** *fosA* **	**x**	**x**	**x**	**x**	**x**	**x**	**x**	**x**	**x**	**x**	**x**	**x**	**x**	**x**
** Ma **	** * mph(A) * **			** x **		** x **	** x **		** x **			** x **	** x **	** x **	** x **
**Ch**	** *catA1* **			**x**		**x**	**x**		**x**			**x**	**x**	**x**	
	** *cmlA5* **							**x**							
** Ph **	** * oqxA * **	** x **		** x **	** x **	** x **	** x **	** x **	** x **	** x **	** x **	** x **	** x **	** x **	** x **
	** * oqxB/B19/B20 * **	** x **	** x **	** x **	** x **	** x **	** x **	** x **	** x **	** x **	** x **	** x **	** x **	** x **	** x **
**Flq**	** *aac(6′)-Ib-cr5* **							**x**		**x**					
	** *gyrA* ** **D87N**	**x**	**x**		**x**	**x**				**x**	**x**				
	** *gyrA* ** **S83I**			**x**			**x**	**x**	**x**			**x**	**x**	**x**	**x**
	** *gyrA* ** **S83Y**	**x**	**x**		**x**	**x**				**x**	**x**				
	** *oqxA* **	**x**		**x**	**x**	**x**	**x**	**x**	**x**	**x**	**x**	**x**	**x**	**x**	**x**
	** *oqxB/B19/B20* **	**x**	**x**	**x**	**x**	**x**	**x**	**x**	**x**	**x**	**x**	**x**	**x**	**x**	**x**
	** *parC* ** **S80I**	**x**	**x**	**x**	**x**	**x**	**x**	**x**	**x**	**x**	**x**	**x**	**x**	**x**	**x**
	** *qnrS1* **	**x**													
** Rf **	** * arr-2 * **							** x **							
**Sph**	** *sul1* **			**x**		**x**	**x**	**x**	**x**			**x**	**x**	**x**	**x**
	** *sul2* **	**x**						**x**							
** Tr **	** * dfrA12 * **			** x **		** x **	** x **		** x **			** x **	** x **	** x **	** x **
	** * dfrA14 * **	** x **				** x **	** x **	** x **		** x **					
**Te**	** *tet(A)* **							**x**							
	** *tet(D)* **									**x**					

β-L, β-lactams; Ag, aminoglycosides; Gy, glycopeptides; Fos, fosfomycin; Ma, macrolide (erythromycin, telithromycin, tylosin); Ch, chloramphenicol; Ph, phenicol; Flq, fluoroquinolones; Rf, rifampicin; Sph, sulfonamide; Tr, trimethoprim; Te, tetracycline. x confirms the detection of the respective gene. A, amikacin; G, gentamicin; T, tobramycin; H, hygromycin; K, kanamycin; KS, kasugamycin; S, streptomycin; SP, spectinomycin; hR, heteroresistant; m, mutant. The font color of different antibiotic classes with their conferring resistance elements is marked alternatively in blue and black color.

**Table 5 antibiotics-11-01171-t005:** Cumulative colistin resistance determinants.

**Bacterial Strain**	** *mcr* ** **Gene**	** *mgrB* ** **Gene**	** *phoP* ** **Gene**	** *phoQ* ** **Gene**	** *pmrA* ** **Gene**	** *pmrB* ** **Gene**	** *crrA* ** **Gene**	** *crrB* ** **Gene**
**BC1_TM**	Negative *	Gene disruption by ISKpn25 insertion (nucleotides 130–144 deleted)	None	None	A217V^n^	T246A^n^	Negative	Negative
**BC2_BM**	Negative	M27K^d^	L4F^d^	Q426L^d^	A217V^n^	T246A^n^	Negative	Negative
**BC3_TM**	Negative	None **	None	L26Q^d^	None	T246A^n^, R256G^d^	None	C68S^n^, Q296L^n^
**BC4_BM**	Negative	M27K^d^	None	Q426L^d^	A217V^n^	T246A^n^	Negative	Negative
**BC5_TM**	Negative	M27K^d^	None	None	A217V^n^	T246A^n^	Negative	Negative
**BC6_BM**	Negative	Gene disruption by ISKpn25 insertion (nucleotides 1–5 deleted)	None	None	None	T246A^n^, R256G^d^	None	C68S^n^
**BC7_BM**	Negative	None	None	None	A41T^n^	P95L^d^, L213M^n^, T246A^n^	None	C68S^n^
**BC8_BM**	Negative	None	None	None	None	T157P^d^, T246A^n^, R256G^d^, V352E^d^	None	C68S^n^, Q296L^n^
**BC9_TM**	Negative	Gene disruption by ISKpn26 insertion (nucleotides 74–144 deleted)	None	None	A217V^n^	T246A^n^	Negative	Negative
**BC10_TM**	Negative	M27K^d^	None	Q426L^d^	A217V^n^	T246A^n^	Negative	Negative
**BC11_TM_B_hR**	Negative	Gene disruption by ISKpn25 insertion (nucleotides 1–5 deleted)	None	L224Q^d^	None	T246A^n^, R256G^d^	None	C68Sn
**BC12_TM_B_m**	Negative	Gene disruption by ISKpn25 insertion (nucleotides 1–5 deleted)	None	L224Q^d^, Q317K^d^	None	T246A^n^, R256G^d^	None	C68S^n^
**BC13_TM_C_hR**	Negative	Gene disruption by ISKpn25 insertion (nucleotides 1–5 deleted)	None	L224Q^d^	None	T246A^n^, R256G^d^	None	C68S^n^
**BC14_TM_C_m**	Negative	Gene disruption by ISKpn25 insertion (nucleotides 1–5 deleted)	None	L224Q^d^	None	T246A^n^, R256G^d^	None	C68S^n^, P151S^d^

**Notes:** * Gene was not identified; ** gene identified without modifications. ^d^Deleterious (intolerant) single nucleotide polymorphisms (SNPs), as predicted by bioinformatic tools. ^n^Neutral (tolerant) SNPs, as predicted by bioinformatic tools. hR, heteroresistant; m, mutant.

**Table 6 antibiotics-11-01171-t006:** PROVEAN prediction for all possible deleterious amino acid substitutions detected in PhoP, PhoQ, PmrB, and CrrB proteins.

Protein	Mutation	PROVEAN Score	Prediction
**PhoP**	L4F	−3.5	Deleterious
**PhoQ**	Q426L	−3.6	Deleterious
	L26Q	−4.2	Deleterious
	L224Q	−4.7	Deleterious
	Q317K	−2.7	Deleterious
**PmrB**	R256G	−5.4	Deleterious
	P95L	−9.6	Deleterious
	T157P	−5.7	Deleterious
	V352E	−3.9	Deleterious
**CrrB**	P151S	−8	Deleterious

**Table 7 antibiotics-11-01171-t007:** Serotype, MLST, cgMLST, and plasmid replicons of all isolates.

**Strain ID**	**Cp. Gene**	**Serotype**	**MLST Sequence Type**	**cgMLST Complex Type**	**Plasmid Replicons (% Identity)**
**BC1_TM**	NDM-1	KL17; O1v1	ST101	CT5839	Col440II (99.29%), IncFIB(pQil) (100%), IncFII(K) (97.97%)
**BC2_BM**	OXA-48	KL17; O1v1	ST101	CT5840	ColRNAI (100%), IncFIA(HI1) (98.45%), IncFIB(K) (100%), IncFII(K) (97.97%), IncL (100%), IncR (100%)
**BC3_TM**	KPC-2	KL106; O2v2	ST258	CT382	ColRNAI (100%), IncFIB(K) (100%), IncFII(K) (97.97%), IncX3(100%)
**BC4_BM**	OXA-48	KL17; O1v1	ST101	CT2059	Col440II (99.29%), ColRNAI (100%), IncFIA(HI1) (98.45%), IncFIB(K) (100%), IncFII(K) (97.97%), IncL (100%), IncR (100%)
**BC5_TM**	KPC-2	KL17; O1v1	ST101	CT5848	Col440II (99.29%), ColRNAI (100%), ColpVC (98.45%), IncFIA(HI1) (98.45%), IncFIB(K) (100%), IncFII(K) (97.97%), IncR (100%)
**BC6_BM**	KPC-2	KL112; O2v2	ST147	CT4408	ColRNAI (100%), ColpVC (98.45%), IncFIA(HI1) (97.16%), IncFIB(K) (100%), IncFIB(pQil) (100%), IncFII(K) (97.97%)
**BC7_BM**	NDM-1	KL102; O2v2	ST307	CT5853	IncFIB(K) (98.93%), IncFII(Yp) (99.13%), IncC (100%)
**BC8_BM**	KPC-2	KL106; O2v2	ST258	CT5854	ColRNAI (100%), IncFIB(K) (100%), IncFII(K) (97.97%), IncX3 (100%)
**BC9_TM**	OXA-48	KL17; O1v1	ST101	CT4633	Col440II (99.29%), IncFIA(HI1) (98.45%), IncL (100%), IncR (100%)
**BC10_TM**	OXA-48	KL17; O1v1	ST101	CT5840	Col440II (99.29%), ColRNAI (100%), IncFIA(HI1) (98.45%), IncFIB(K) (100%), IncFII(K) (97.97%), IncL (100%), IncR (100%)
**BC11_TM_B_hR**	KPC-2	KL112; O2v2	ST147	CT4408	ColRNAI (100%), IncFIB(K) (100%), IncFIB (pQIL) (100%), IncFII(K) (97.97%)
**BC12_TM_B_m**	KPC-2	KL112; O2v2	ST147	CT4408	ColRNAI (100%), IncFIB(K) (100%), IncFIB (pQIL) (100%), IncFII(K) (100%)
**BC13_TM_C_hR**	KPC-2	KL112; O2v2	ST147	CT4408	ColRNAI (100%), ColpVC (98.45%), IncFIB(K) (100%), IncFIB (pQIL) (100%), IncFII(K) (97.97%)
**BC14_TM_C_m**	KPC-2	KL112; O2v2	ST147	CT4408	ColRNAI (100%), ColpVC (98.45%), IncFIB(K) (100%), IncFIB (pQIL) (100%), IncFII(K) (97.97%)

Cp, carbapenemase; MLST, multilocus sequence type; cg, core genome; hR, heteroresistant; m, mutant.

**Table 8 antibiotics-11-01171-t008:** Demographic data of patients and general details of CR and ChR *K. pneumoniae* CPE isolates.

Bacterial Isolate	Cp. Type	Source	Date of Collection	Town of Isolation	Hospital Unit	Patient Gender	Patient Age (Years)
**BC1_TM**	MBL	Blood	14.01.2019	Târgu Mureș	Medical	M	63
**BC2_BM**	OXA-48-like	Wound	30.07.2019	Baia Mare	Medical	M	79
**BC3_TM**	KPC	Tracheal aspirate	01.02.2018	Târgu Mureș	ICU	M	66
**BC4_BM**	OXA-48-like	Urine	11.06.2018	Baia Mare	Surgical	M	70
**BC5_TM**	KPC	Wound	09.08.2018	Târgu Mureș	ICU	M	26
**BC6_BM**	KPC	Urine	23.06.2020	Baia Mare	Medical	M	75
**BC7_BM**	MBL	Urine	23.04.2021	Baia Mare	Surgical	F	68
**BC8_BM**	KPC	Blood	21.11.2017	Baia Mare	Surgical	M	67
**BC9_TM**	OXA-48-like	Blood	28.01.2019	Târgu Mureș	ICU	F	91
**BC10_TM**	OXA-48-like	Tracheal aspirate	08.01.2019	Târgu Mureș	ICU	M	68
**BC11_TM_B_hR**	KPC	Tracheal aspirate	28.03.2019	Târgu Mureș	ICU	M	45
**BC12_TM_B_m**	KPC	-	-	-	-	-	-
**BC13_TM_C_hR**	KPC	Tracheal aspirate	04.03.2019	Târgu Mureș	ICU	F	64
**BC14_TM_C_m**	KPC	-	-	-	-	-	-

Cp, carbapenemase; MBL, metallo-β-lactamase; OXA-48-like, oxacillinase OXA-48-like; KPC, *K. pneumoniae* carbapenemase; ICU, intensive care unit; hR, heteroresistant; m, mutant.

## Data Availability

Data are contained within the article, and all the raw reads are available in the ENA database.

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
