# Peer review of "Characterization of Carbapenemase-Producing Klebsiella pneumoniae Isolates from Two Romanian Hospitals Co-Presenting Resistance and Heteroresistance to Colistin"

_antibiotics, 2022, doi:10.3390/antibiotics11091171_

Round 1
Reviewer 1 Report
In the manuscript titled " Characterization of carbapenemase-producing Klebsiella pneumoniae isolates co-presenting resistance or heteroresistance to colistin from two Romanian hospitals"
The authors selected clinical strains of Klebsiella from patient samples and performed antibiotic susceptibility testes to identify the resistance spectrum in the strains and have further used WGS to get a understanding about the prevalence of different resistance phenotypes within the strain.
I have few suggestions that can further improve the reading and delivery of the scientific content of the study.
1)The introduction and discussion sections are rather long. Shortening and making these sections more focused will help the reader appreciate the work.
2) Few typographical and language corrections are needed to improve the quality of the manuscript.
3) The list of references could be reduced (more than 100 seems way too much for a study of this type) and could be reduced to include the most appropriate ones only.
Author Response
We are very grateful to the Referee 1 for the precious time in reviewing our paper and offering the great opportunity to improve the manuscript and making it more attractive. We have been able to incorporate changes to reflect the suggestions provided and have used the Track Changes function.
Here is a point-by-point response to the reviewers’ comments and concerns.
Comment 1: The introduction and discussion sections are rather long. Shortening and making these sections more focused will help the reader appreciate the work.
Response 1: Thank you for pointing this out. We have deleted unnecessary information presented in phrases and paragraphs from introduction (lines 54-55, 76-77, 81-84, 110-111), results (lines 186, 201-204), and discussion sections (lines 364-365, 367-371, 382-387, 441-446, 521-523, 542-551), in association with minor reformulations (lines 85, 364-365, 388). In results section rows 129-136 we have reintroduced and deleted a paragraph which was present in the original manuscript submitted when all pages were in portrait orientation layout. In row 319 we have introduced in text Figure 1 and have renamed Table S5 into S1 in line 207. The numbering of the tables 6,7,8 and 9 has been modified accordingly. In line 121 the term “successful plasmids” was replaced with “dominant plasmids”. Additionaly, the numbering of the bibliographic sources has been corrected accordingly in all manuscript. In line 807 Data Availability Statement has been completed as following: “Data are contained within the article and all raw reads are available in ENA”.
Comment 2: Few typographical and language corrections are needed to improve the quality of the manuscript.
Response 2: Thank you for this observation. We will apply for a specialist English language editing service provided by MDPI when will submit the revised manuscript.
Comment 3: The list of references could be reduced (more than 100 seems way too much for a study of this type) and could be reduced to include the most appropriate ones only.
Response 3: We agree with this and have incorporated your suggestion. The list of reference has been reduced to 99 essential sources. All online available sources have been reaccessed on 10th August 2022.
English language and style: (x) English language and style are fine/minor spell check required
Response: please accept the response to comment 2.
Reviewer 2 Report
Brief Summary
FÅ‘ldes et al. embark on a journey to measure the current antibiotic resistance in clinical isolates collected from 2 Romanian hospitals. Specifically, they measure the Colistin resistance in Klebsiella pneumonia. FÅ‘ldes et al. discuss how the new data compares to data from other hospitals in Romania today and in the past.
Significance
The emergence of colistin resistance in clinical isolates constitutes a troubling trend at hospitals in Romania and around the world. Studies describing the current resistant isolates are urgently needed. This type of data will be used to better understand the challenges that we are facing today and to combat rising antibiotic resistance in the future.
Recommendations:
I enthusiastically recommend accepting this paper with minor revisions for publication at the Antibiotics Journal. I am listing below minor suggestions for clarifying details described in this review. I am not recommending any additional experiments.
Notes on the text:
General:
- This paper is very long. This paper can benefit from keeping the results and discussion more concise, which will help the readers stay focused on the results and their meaning and impact. I suggest shortening some of the discussion in the results and the discussion sections. This great extra discussion and review of the filed could be featured in a separate review on the topic.
- Figure numbering is not consistent. Please rename table S5 to table S1, and figure 2 to figure 1.
Line 119: Is “successful” the best word choice? I suggest using another descriptor, such as “permanent” or “dominant” in this sentence.
Author Response
We are very grateful to the Referee 2 for the precious time in reviewing our paper and offering the great opportunity to improve the manuscript and making it more attractive. We have been able to incorporate changes to reflect the suggestions provided and have used the Track Changes function.
Here is a point-by-point response to the reviewers’ comments and concerns.
Comment 1: This paper is very long. This paper can benefit from keeping the results and discussion more concise, which will help the readers stay focused on the results and their meaning and impact. I suggest shortening some of the discussion in the results and the discussion sections. This great extra discussion and review of the filed could be featured in a separate review on the topic.
Response 1: Thank you for pointing this out. We have deleted unnecessary information presented in phrases and paragraphs from introduction (lines 54-55, 76-77, 81-84, 110-111), results (lines 186, 201-204), and discussion (lines 364-365, 367-371, 382-387, 441-446, 521-523, 542-551) sections, in association with minor reformulations (lines 85, 364-365, 388). In results section rows 129-136 we have reintroduced and deleted a paragraph which was present in the original manuscript submitted when all pages were in portrait orientation layout. In row 319 we have introduced in text Figure 1. Additionaly, the numbering of the bibliographic sources has been corrected accordingly in all manuscript. In line 807 Data Availability Statement has been completed as following: “Data are contained within the article and all raw reads are available in ENA”.
Comment 2: Figure numbering is not consistent. Please rename table S5 to table S1, and figure 2 to figure 1
Response 2: We agree with this and have incorporated your suggestions. The numbering of the tables 6,7,8 and 9 has been modified accordingly.
Comment 3: Line 119: Is “successful” the best word choice? I suggest using another descriptor, such as “permanent” or “dominant” in this sentence.
Response 3: Thank you for this observation. In the new line 121 (equivalent to mentioned line 119) the term “successful plasmids” was replaced with “dominant plasmids”.
English language and style: (x) English language and style are fine/minor spell check required
Response: We will apply for a specialist English language editing service provided by MDPI when will submit the revised manuscript.
